# Spatiotemporal Features—Extracted Travel Time Prediction Leveraging Deep-Learning-Enabled Graph Convolutional Neural Network Model

**Xiantong Li** [1], **Hua Wang** [1,*], **Pengcheng Sun** [1] **and Hongquan Zu** [2]

[1] School of Transportation Science and Engineering, Harbin Institute of Technology, Harbin 150001, China; lxt@hit.edu.cn (X.L.); 18S132096@stu.hit.edu.cn (P.S.)

[2] Network Center, Harbin Institute of Technology, Harbin 150001, China; zuhq@hit.edu.cn

[*] Correspondence: wanghua@hit.edu.cn; Tel.: +86-138-0458-4448

**Abstract:** Travel time prediction is one of the most important parameters to forecast network-wide traffic conditions. Travelers can access traffic roadway networks and arrive in their destinations at the lowest costs guided by accurate travel time estimation on alternative routes. In this study, we propose a long short-term memory (LSTM)-based deep learning model, deep learning on spatiotemporal features with Convolution Neural Network (DLSF-CNN), to extract the spatial–temporal correlation of travel time on different routes to accurately predict route travel time. Specifically, this model utilizes network-wide travel time, considering its topological structure as inputs, and combines convolutional neural network and LSTM techniques to accurately predict travel time. In addition to their spatial dependence, both coarse-grained and fine-grained temporal dependences are fully considered among the road segments along a route as well. The shift problem is formulated in the coarse-grained granularity to predict the route travel time in the next time interval. The experimental tests were conducted using real route travel time obtained by taxi trajectories in Harbin. The test results show that the travel time prediction accuracy of DLSF-CNN is above 90%. Meanwhile, the proposed model outperformed the other machine learning models based on multiple evaluation criteria. The RMSE (Root Mean Square Error) and $R^2$ (R Squared) increased by 18.6% and 22.46%, respectively. The results indicate the proposed model performs reasonably well under prevailing traffic conditions.

**Keywords:** spatial–temporal feature; convolutional neural network; long short-term memory network; attention mechanism

## 1. Introduction

During the past two decades, traffic congestion has become increasingly serious in metropolitan urban areas. Effectively monitoring traffic conditions is essential for optimal traffic management and control strategies. Travel time is one of the most important parameters to describe traffic operation conditions and provide important information to travelers for their trip planning and en-route guidance. Real-time and accurate travel time can effectively reflect the traffic conditions in roadway networks. Therefore, it affects the route selection of a traveler to get to their destination at the lowest cost. The great improvements in traffic management and control technology have enhanced traffic information collection system performance in the field, so that it is possible to measure real-time traffic data for travel time prediction. For example, autonomous vehicle identification (AVI) systems, automatic license plate recognition cameras, Bluetooth devices, etc., can accurately measure route travel time directly. However, due to the limited coverage of fixed-location traffic detectors, it is difficult to obtain traffic data for entire roadway networks. Under such situations, floating car data extracted from geographic position system (GPS) devices installed in vehicles or mobile phones are more and more attractive. Floating car data collection

technology, as a means of mobile information collection, utilizes floating car trajectories for corridor-wide travel time collection at low costs with high data accuracy. Floating car data, which can be divided into high- or low-resolution parts, can provide mobile traffic information for the entire network to supplement fixed-location sensor outputs. For instance, Yuan et al. evaluated the main line traffic situation through floating car data and loop coil detector data in [1]. Liu et al. use the concept of virtual inspection cars to infer travel time based on signal status data from detectors along with corridors in [2]. However, due to the sparse nature of floating car trajectories, vehicles may cross multiple road segments between adjacent tracks. It is challenging to predict the travel time of a road segment based on floating car trajectory data only under certain conditions, especially on low-resolution datasets.

Travel time prediction requires correlation discovery in spatiotemporal dimensions and has been studied widely. For example, the moving average model and its variant methods have been usually used in the early stage to predict travel time using time series modeling techniques, such as differential autoregressive moving average model in [3]. Based on this model, other feature extraction and machine learning models are used for time series prediction problems. However, these traditional time series models have certain limitations. A major problem is that they cannot capture the dependence among data in time and space series.

In order to discover and quantify the data spatiotemporal dependence of, and better predict, route-wide travel time, in this study, we aim to model the spatial and temporal dependence of travel time data separately based on spatiotemporal feature-enabled deep learning models combined with travel time data characteristics. The main contributions of this paper are as follows.

(1)　A convolutional neural network is used to capture the spatial dependence among a series of partial paths included in a route.

(2)　An long short-term memory (LSTM) network is used to capture the time-sequence dependence of travel time.

(3)　An attention mechanism is used to correct the systematical errors from the periodic drift of travel time data to the long-term sequence to realize accurate route travel time prediction. The convolutional neural network and LSTM network are combined, becoming the model of DLSF-CNN (deep learning on spatiotemporal features with Convolution Neural Network, CNN).

However, the model in this paper is proposed to focus on low-resolution datasets from floating cars. This is also the limitation of DLSF-CNN. Considering the complexity of the algorithm on this model, it is not efficient enough to handle high-resolution datasets. The future study plan is to design a model similar to DLSF-CNN, which is simpler but efficient, to realize the estimations on high-resolution datasets.

The notations used in this paper are shown in Appendix A and acronyms in this paper are shown in Appendix B at the end of this paper.

The rest of this paper is organized as follows. Related work is reviewed in Section 2. In Section 3, the formal problem definition is given. The route travel time prediction method and its performance evaluation are given in Sections 4 and 5 separately. The discussion of this effort is given in Section 6.

## 2. Related Work

Deep learning is a powerful branch of artificial intelligence (AI) research. It has shown favorable results with its unique learning ability in many fields, such as computer vision, natural language processing, etc., [4]. Recently, more and more researchers introduced deep learning methods to the field of transportation science and engineering. For example, convolutional neural network captures the spatial correlation among data by continuously performing convolution operations on the elements in the matrix [5]. Recurrent neural network discovers time correlation by modeling time series data [6]. These two methods can be combined to form a complicated deep learning model. This model overcomes

the limitations of each model and has the ability to process multidimensional data with spatiotemporal characteristics.

Due to the spatiotemporal characteristics of route travel time, travel time prediction is similar to other traffic parameter estimations. The common prediction goal is to estimate the relevant index value of a certain parameter in a certain time interval, such as traffic flow prediction, traffic speed prediction, etc., based on their similar data characteristics. Therefore, the related research in the other field can be used as a benchmark for travel time prediction. The prediction approaches can be roughly divided into two categories: prediction models based on statistical methods and prediction models based on deep learning.

The statistical approach-based prediction model mainly uses the classic time series analysis method to estimate unknown data in the temporal dimension. The preferred result of the prediction is obtained by statistically extracting data correlations and similarities from different time periods. Furthermore, traditional traffic flow prediction methods can be divided into parametric methods and non-parametric methods. Parametric methods include differential autoregressive moving average models and its variant methods [7,8] (DAMAMV). Non-parametric methods include K-nearest neighbor regression method [9], historical average method, vector autoregression method [10] and so on. DAMAMV methods rely on uninterrupted input sequences that are not suitable for analyzing time series with missing data. The historical average model cannot effectively capture the dynamic changes of traffic data, such as periodic features. Vector autoregressive models can capture the linear correlation among related time series, but then ignore the correlation among predicted values. All these methods only consider the temporal characteristics, and they cannot capture the complex non-linear spatiotemporal dependence among data.

Deep learning-based prediction models capture the spatial dependence within data by handling traffic data prediction as an image matrix. Treiber et al. [11] estimated travel time of urban arterials based on a neural network model. This method constructs traffic flow conditions along road segments through single-segment models in neural networks. If a route is considered, it should combine multiple single-segment models. It can reduce the number of neural network input parameters and improve the operating efficiency of the system. Li et al. [12] proposed a traffic flow prediction method based on residual LSTM networks. They used integrated learning approaches to train spatially distributed data end-to-end in a residual LSTM network. At the same time, a dimension weighting unit is introduced behind every LSTM unit. The interdependence between feature dimensions is modeled explicitly. Zhang et al. [13] captured the spatial dependence through convolutional neural network models. These use the residual neural network to predict urban people transfer flow. Yu et al. [14] used a recurrent neural network model to model the dependence of the time characteristics of variables. Though these studies are based on temporal dependence or spatial dependence, they do not consider the dependence between these two dimensions simultaneously.

A few recent studies were conducted to use CNN and LSTM simultaneously to capture the spatial and temporal dependence among the data. Yao [15,16] used local CNN, LSTM, and a semantic graph embedding to capture the spatial, temporal, and semantic dependence, respectively. Their algorithm predicted the regional taxi demands from multiple perspectives. Guo [17] extracted the spatial features of the data and input them into an LSTM network and deployed the attention mechanism to enhance the weight of the representative data influence on each prediction step. Shah [18] and Iqbal [19] used machine learning methods along with logistic regression, support vector machine, and CNN in cancer cell classification; they improved the accuracy in relation to diseased cell identification to 98%. However, in travel time prediction, the complex nonlinear temporal and spatial dependence of the partial and complete paths included in a route significantly affects its prediction accuracy. Very few studies have fully considered the spatial and temporal dependence. Therefore, we aim to use partial coverage path travel time data to construct a route travel time matrix and propose a deep learning model based on spatial–

temporal features to deal with the dependence between complex spatial and temporal dimensions to predict route travel time.

## 3. Materials and Methods

To extract the spatial dependence among data by using convolutional neural network models, it is necessary to pre-process travel time data. The spatiotemporal data represented by the grid matrix should be converted into the image-type data. In addition, travel time data demonstrate obvious temporal correlation. A more accurate model capable of describing temporal dependence among data can combine short-term sequences of recent timeslots and long-term sequences of past days or even months. Some key definitions are listed as follows to facilitate our formulation process.

**Definition 1** (**Route**). *Route denotes an entire traffic path from the start point to the end point in a roadway network, where the origin or the destination of the path is an intersection. A set of roadway segments included in a route is $L = \{L_1, L_2, L_3, \cdots, L_n\}$, where n is the number of roadway segments along this route.*

**Definition 2** (**Timeslots**). *A day is divided into t slices according to the interval of a slice, namely, $T_0, T_1, T_2, T_3, \cdots, T_{t-1}$. When the internal of each time slice is 15 min, it is called a timeslot.*

**Definition 3** (**Travel Time Samples**). *When the i-th timeslot is considered, the travel time samples in a route can be represented by Vector $t_{ijk}$, that is $t_{ijk} = (T_i, O_j, D_k)(i = \{1, \ldots, t\}, j, k = \{1, \ldots, n\}, j < k)$, where $T_i$ represents the i-th timeslot, and $O_j$ and $D_k$ represent the j-th and k-th roadway segments of the origin and destination point located, respectively.*

According to Definition 3, $t_{ijk}$ records the travel time taken by a vehicle from the $j$-th road segment to the $k$-th road segment in the $i$-th timeslot.

**Definition 4** (**Timeslot Travel Time**). *Timeslot travel time is represented by $\overline{t_{ijk}}$. It is calculated through weighted average methods on all travel time samples under the i-th timeslot. For example, if there are m travel time samples in total under the i-th timeslot, the calculation formula of timeslot travel time is shown in Equation (1).*

$$\overline{t_{ijk}} = \frac{1}{m}\sum_{p=1}^{m} t_{ijk}, p \tag{1}$$

Here, $p$ represents one of the $m$ samples.

**Definition 5** (**Route Travel Time Matrix**). *When the i-th timeslot is considered, the route travel time matrix is represented by $R_i$, as shown in Equation (2).*

$$R_t = \begin{bmatrix} \overline{t_{i,1,1}} & \cdots & \overline{t_{i,1,n}} \\ \vdots & \ddots & \vdots \\ \overline{t_{i,n,1}} & \cdots & \overline{t_{i,n,n}} \end{bmatrix} \tag{2}$$

The size of the matrix is $n \times n$, where $n$ is the number of paths included in a route. The elements in the matrix are the travel time spent at the corresponding locations as part of the route. The travel time matrix not only reflects the relationship between travel time of a route and its roadway segments, but also records the relationship among travel time in different directions.

The research goal in this study, known as Route Travel Time Prediction, is to predict the route travel time at the $(t + 1)$-th time stamp when the $t$-th route travel times from 1 to $t$ intervals are given in the route travel time matrix.

## 4. Deep Learning Prediction Model Based on Spatiotemporal Features

The proposed spatiotemporal feature-based deep learning prediction model uses convolutional neural networks to capture the spatial dependence of a route. It also uses LSTM networks to extract temporal dependence of travel time. Due to the time shift in LSTM networks, the attention mechanism is introduced to correct such drift. Figure 1 shows the overall architecture of this model.

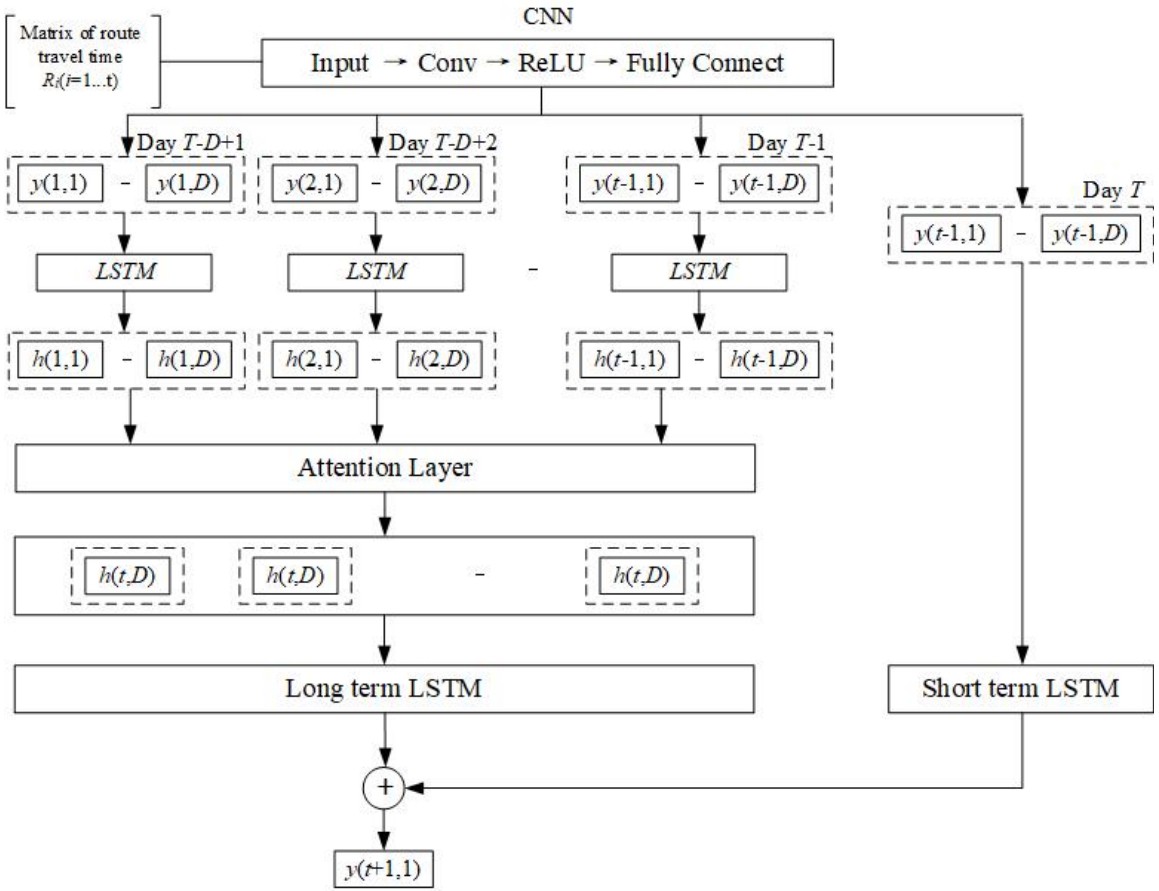

**Figure 1.** This is the structure of the model DLSF. It contains modules such as CNN, temporal modules, attention layer, long-term LSTM (long short-term memory), and short-term LSTM. It forecasts time slot $t + 1$ on the calculation from time slots 1 to $t$.

### 4.1. Spatial Characteristics

A spectral domain graph convolution network is used in this model. It realizes the convolution operation on the topological graph by the spectral graph theory. The spectral domain-based graph convolution extends the grid-based convolution to graph. It transforms graph data into frequency domain inner product by time domain convolution kernel $g$ and input signal $x$. The final filtering process in frequency domains is conducted by calculating the Laplacian matrix and its eigenvalues. In this paper, floating car data are modeled based on the spectrogram theory. The correlation relationship among urban traffic network road segments is calculated in spatial dimensions for the purpose of describing floating car trajectory accurately.

The proposed method extracts the spatial correlation from the pixel matrix through a convolutional neural network to capture the spatial dependence of travel time in routes. First, it models every partial path into a complete route by every corresponding route travel time matrices in a single timeslot. Second, it sets the matrix as a grayscale image to input into a convolutional neural network for spatial feature relationship learning. At last,

the route travel time matrix is developed as an input sample of the vertex in a timeslot. If there are no travel time data on a partial path of a route, the free-flow travel time is used instead. The convolutional neural network extracts original $R_t^i$ as $R_t^i, 0$ to input it into the convolutional layer *k*. Two-dimensional convolution is used to extract the spatial characteristics of the route travel time. The convolution formula is shown in Equation (3).

$$R_t^i, \mathrm{k} = \mathrm{ReLu}\left(\omega_k * R_t^i, \mathrm{k} - 1 + \mathrm{b}\right) \tag{3}$$

where *k* is the number of the convolution layers, $*$ is a convolution operator, ReLu is an activation function, $\omega_k$ is a weight coefficient, and *b* is a constant. After performing the convolution operation, the fully connected layer is used to process the extracted spatial correlation information into the input $R_t^i$ of LSTM.

### 4.2. Attention-Mechanism-Based LSTM Network

### 4.2.1. LSTM Network Principle

The LSTM network is a prevalent application of recurrent neural networks. Its basic unit is shown in Figure 2, which includes input gate, forget gate, and output gate. The LSTM unit calculates the hidden layer output $h_t$ at time *t* through the input value $x_t$, as well as the memory cell state value $c_{t-1}$ and hidden layer output value $h_{t-1}$ at time *t*-1. According to the sequence flow, the formula for calculating the forgetting layer is shown in Equation (4). The forgetting layer determines the information to be forgotten to predict a certain effect.

$$f_t = \sigma\left(W_f \cdot [h_{t-1}, x_t] + b_f\right) \tag{4}$$

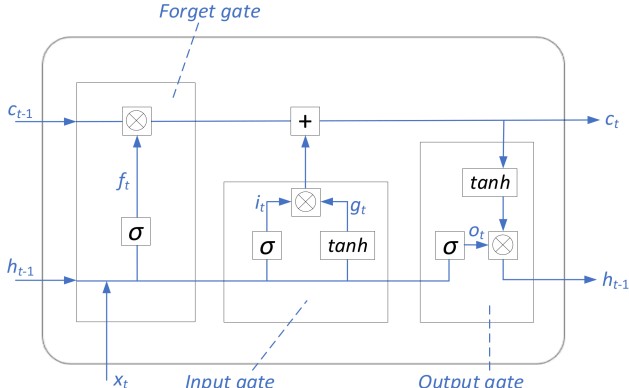

**Figure 2.** This is the basic structure of unit LSTM. The main function of this unit is to calculate the hidden layer in a neural network.

The calculation formula of the input layer is shown in Equations (5) and (6). Equation (5) determines the information to be updated, and Equation (6) is used to update the content.

$$i_t = \sigma(W_i \cdot [h_{t-1}, x_t] + b_i) \tag{5}$$

$$C_t' = tanh(W_C \cdot [h_{t-1}, x_t] + b_C) \tag{6}$$

The forget gate and the input gate have decided which information should be kept and forgotten. If the two parts are combined together, it is much more precise to renew the information. The combination is shown in Equation (7).

$$C_t = f_t * C_{t-1} + i_t * C_t' \tag{7}$$

The output information is drawn with the decision made by the sigmoid layer, which is shown in Equations (8) and (9).

$$o_t = \sigma(W_o \cdot [h_{t-1}, x_t] + b_o) \tag{8}$$

$$h_t = tanh(C_t) * o_t \tag{9}$$

where $W$ is the weight matrix at time $t$, $b$ is the bias constant, $\sigma(\cdot)$ is the sigmoid activation function, and $tanh(\cdot)$ is the tanh activation function.

### 4.2.2. Fine Particle Temporal Characteristics

The LSTM network recursively applies the transfer function to the hidden state vector, which solves the problem of gradient explosion or disappearance in the ordinary recurrent neural network. The LSTM network shows promising performance for time series data modeling. Therefore, the model predicts the result of the $(t + 1)$-th timeslot by calculating $t$ previous short-term time series, while it uses a fine particle temporal short-term sequence based on the LSTM learning path travel time. The LSTM network prediction can be expressed as Equation (10).

$$h_t^i = LSTM\left(R_t^i, h_{t-1}^i\right) \tag{10}$$

where $h_t^i$ is the prediction result of the partial path $i$ in a route in the $t$-th timeslot, and $R_t^i$ is the output of the convolutional neural network.

### 4.2.3. Coarse Granularity Temporal Characteristics

The LSTM network only considers the fine-grain time characteristics of the first few timeslots of the route travel time, but it does not consider the periodical coarse-grain time series characteristics of the travel time. If the input length of the time series increases, the LSTM ability to capture long-term time series features decreases. Therefore, deep learning on spatiotemporal features with CNN (DLSF-CNN) uses the input data from the same number of timeslots of the past $P$ days for the long-term LSTM network. However, due to systematical time shift in periodic travel time data, using long-term sequential travel time data as the only input into the long-term LSTM network will produce the considerable deviation of the prediction results.

The DLSF-CNN introduces an attention mechanism to correct the influence of the time shift based on the LSTM network and finally obtains an LSTM model based on the attention mechanism. As shown in Figure 1, the historical route travel time data in the $D$-th timeslot in the past $P$ days are used as the input of the long-term LSTM model. For example, the route travel time in the timeslot from 9:45 to 10:00 a.m. on a certain day is predicted, and the data in the two hours before that timeslot are extracted as input. In this case, $D = 8$ when the timeslot interval is 15 min. Extracting the time series travel time data of each day in the past P days by the LSTM network is used to learn the weights of different timeslots to the final vector representation of the day to correct the systematical errors caused by the time shift. The formula based on the attention mechanism and weight distribution can be expressed in the following equations.

$$h_d^{i,p} = LSTM\left(R_d^{i,p}, h_{d-1}^{i,p}\right) \tag{11}$$

$$s = v * \tanh\left(\omega_H h_d^{i,p} + \omega_X h_t^i + b\right) \tag{12}$$

$$\alpha_d^{i,p} = \frac{exp(s)}{\sum_{d \in D} exp(s)} \tag{13}$$

$$h_t^{i,p} = \sum_{d \in D} \alpha_d^{i,p} * h_d^{i,p} \tag{14}$$

where $R_d^{i,p}$ is the output of the convolutional neural network on the travel time along route $i$ in the $d$-th timeslot of the $p$-th day. $v$, $\omega_H$, $\omega_X$, and $b$ are the learning parameters of the model. $s$ is the contribution score function. $\alpha_d^{i,p}$ is the weight of the attention contribution. $h_d^{i,p}$ is the vector representation of the travel time of route $i$ in the $d$-th timeslot on the $p$-th day. $h_t^{i,p}$ is the prediction of the travel time of route $i$ in the $t$-th timeslot on the $p$-th day. The final vector representation of the long-term LSTM prediction model can be expressed as Equation (15).

$$h'^{i,p}_t = LSTM\left(h_t^{i,p}, h'^{i,p-1}_{t-1}\right) \tag{15}$$

where $h'^{i,p}_t$ is the prediction result based on the time characteristics of coarse granularity with time shift correction by the attention-mechanism-based LSTM network.

*4.3. Fusion Model*

The DLSF-CNN integrates the output results from the short-term LSTM model and long-term LSTM model to obtain $h_t^{i,l}$. Then, the activation function is deployed to compute the prediction of the $i$-th route travel time in the $(t + 1)$-th timeslot. The calculation formula of the prediction model is in Equation (16).

$$y_{t+1}^i = \tan h\left(\omega_l h_t^{i,l} + b\right) \tag{16}$$

where $\omega_l$ and $b$ are the learning parameters of the model, and $y_{t+1}^i$ is in the range of $(-1,1)$, as the span of function *tanh*(). When the prediction result is denormalized, the predicted value $y_{t+1}^i$ of route $i$ in the $(t + 1)$-th timeslot is obtained.

## 5. Experimental Test Results

*5.1. Experimental Environment*

Experimental tests were conducted using the 5000 taxis' trajectory data collected by the Harbin Transportation Bureau. The data include the vehicle equipment number, location (longitude, latitude), time stamp, and other status information every minute on average. The applicability of the data from these floating cars and the action track coverage area are shown in Figure 3. It is the urban traffic network of Harbin, China. Five hundred taxies are selected to study the route travel time from 5000 taxies in total. The ground-truth value of one route travel time is the actual time spent by a floating car so that we compare the predicted travel time of each real route and the accuracy of the model can be calculated. The frequency of the data collection of a floating car is 30 s, though it is a low-resolution dataset to be effectively studied for each equivalent problem. However, it is still worth careful study as the low-frequency floating car data represent a big proportion of all such datasets.

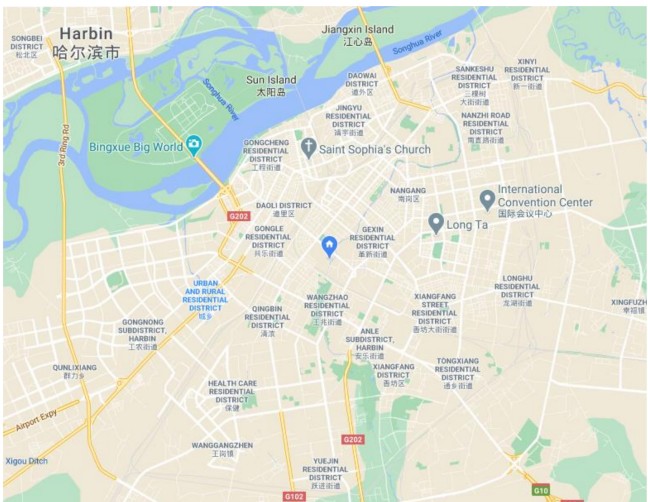

**Figure 3.** The main area of experimental data in this paper where the study took place.

The low-frequency floating car data need to be preprocessed before they can be used to estimate the route travel time. It is fundamentally important to match the floating car trajectory to the road network and perform path inference. Many previous studies [20–22] developed different map matching and path reasoning methods. To simplify our travel time prediction analysis, we assume that the floating car data have been fully processed based on map matching and path reasoning.

The taxi trajectory data are utilized from 7:00 a.m. To 9:00 a.m. between 17 December 2016 and 17 January 2017. The data are divided into a timeslot of 15 min and 7:00–7:15 a.m. is the first timeslot of a day. There are eight timeslots of a day in the selected 2 h. The research timeslot is selected as the eighth timeslot on 17 January 2017, from 8:45 to 9:00 a.m. Five hundred taxis were chosen randomly in this timeslot and 70% of them are selected for training purposes and 30% for model validation.

All data analyses were performed in a 64-bit Microsoft Windows 7 operating system on Intel(R) Core (TM) i5−4210U CPU @ 1.70GHz 2.40 GHz, 4 GB memory and 1 T mechanical hard drive. The programming code was written based on Python3.7 platforms, and the software architecture is developed based on Keras deep learning library tools.

*5.2. Evaluation Index*

The average absolute error and $R^2$ score are chosen to evaluate the prediction results of the model.

(1) Mean absolute error

$$\text{MAE} = \frac{1}{m} \sum_{i=1}^{m} |\hat{y}_i - y_i| \tag{17}$$

(2) R-squared

$$R^2 = 1 - \frac{\sum_{i=1}^{m} (\hat{y}_i - y_i)^2}{\sum_{i=1}^{m} (\overline{y}_i - y_i)^2} \tag{18}$$

*5.3. Result Analysis*

The DLSF model is a deep learning network model to predict travel time by extracting their spatiotemporal features. The DLSF-CNN performance is evaluated based on the taxi trajectory dataset and compared with the Differential Autoregressive Moving Average (DAMA) model [3], the one-way LSTM [12] network model, and the two-way LSTM [17] network model. Table 1 shows the estimation results of the different models.

**Table 1.** Comparison between different models.

| Model | MAE | $R^2$ |
|---|---|---|
| DAMA | 28.22 | 0.641 |
| One-way LSTM | 24.34 | 0.711 |
| Two-way LSTM | 24.48 | 0.719 |
| DLSF-CNN | 22.97 | 0.785 |

Table 1 shows that the one-way LSTM model and the two-way LSTM model produced the same predictive performance on route travel time. Compared with the DAMA model, the average absolute error and $R^2$ scores are improved. However, the DLSF-CNN model outperformed in the prediction accuracy over the other three models. These results illustrate that the predictive ability of the DLSF-CNN model is better than the other methods. Compared with the DAMA model, the average absolute error and $R^2$ score are improved by 18.6% and 22.46%, respectively.

In the deep learning process, the expansion of components can improve the algorithm accuracy, but the execution efficiency of the algorithm will be impacted along with such expansion. Table 2 shows the comparison results of the algorithm efficiency after deleting unusable components in the model.

**Table 2.** Efficiency comparison between different algorithms from different models.

| Model | MAE | $R^2$ |
|---|---|---|
| DAMA, removed CNN, ignoring spatial dependencies | 26.83 | 0.711 |
| One-way LSTM, removed short-term LSTM network, using CNN and long-term LSTM | 24.42 | 0.725 |
| Two-way LSTM, removed long-term LSTM network, using CNN and short-term LSTM | 23.96 | 0.738 |
| DLSF-CNN | 22.69 | 0.762 |

Table 2 shows when some of the components in a model are removed, the overall prediction performance of the model decreases. In addition, when the short-term LSTM network is replaced by CNN and long-term LSTM networks, the prediction performance degrades compared to the scenario that the long-term LSTM network is replaced by CNN and short-term LSTM networks. Through this analysis, it can be found that the short-term time series can capture the time dependence among the data instead of the long-term time series. This result is consistent with the theoretical analysis result.

Figure 4 shows the prediction comparison with the ground-truth data. The prediction results are estimated through the DLSF-CNN algorithm, and the ground-truth travel time in a certain timeslot of 50 paths are randomly selected from the 500 paths shown by the green curve. The results show that the prediction accuracy of the DLSF-CNN algorithm is about 90%, and the average relative errors are less than 4.66%. This result also shows that the proposed route travel time prediction method can accurately predict the path travel time reasonably.

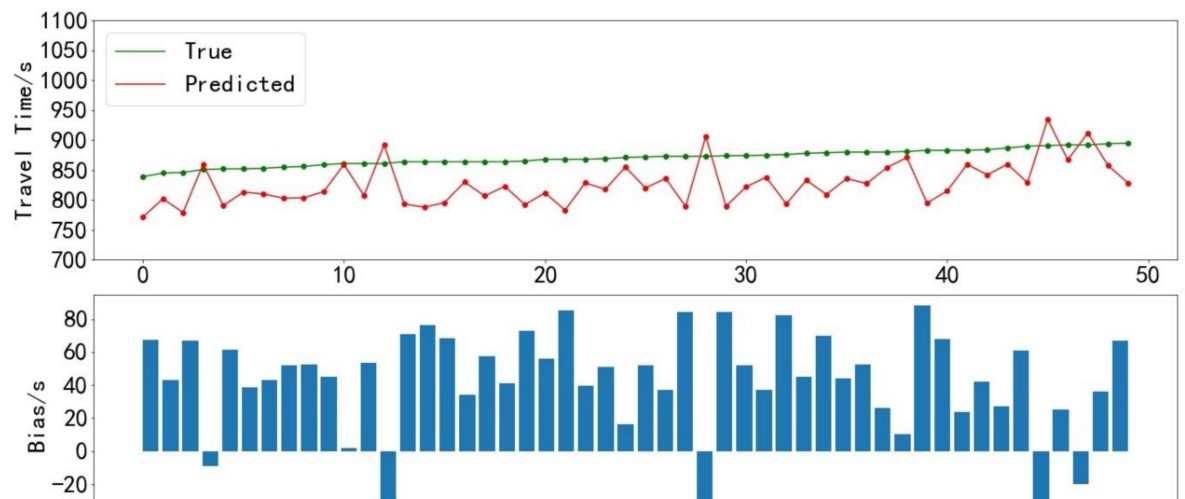

We also compared the prediction results between DLSF-CNN and the other algorithms on the 50 selected routes to further verify the performance of each algorithm. The comparison results are shown in Figure 5. The ground-truth data are illustrated by the green and gentle curve. As we can see, the prediction abilities of the one-way LSTM model and two-way LSTM model are basically the same. They both outperformed the DAMA model, although the DLSF-CNN algorithm performed best. This result is consistent with the experiment's result in terms of the error tests.

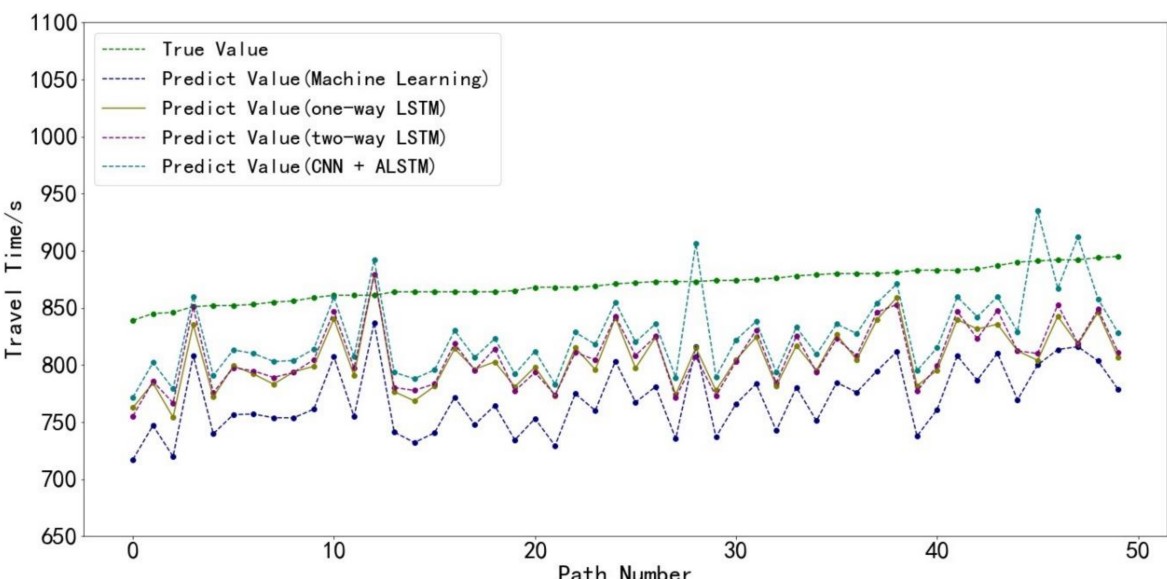

**Figure 5.** These results are the comparison on 4 predictions from the corresponding route travel time models to ground-truth data.

To further verify the effectiveness of the DLSF-CNN algorithm, four routes are selected randomly to conduct additional experiments during a whole day. The average travel time estimation results of every timeslot in a day are shown in Figures 6 and 7 through the line chart and box plot, respectively. In this experiment, the DLSF-CNN algorithm describes the change patterns accurately in travel time prediction, especially during rush hours in the morning and evening.

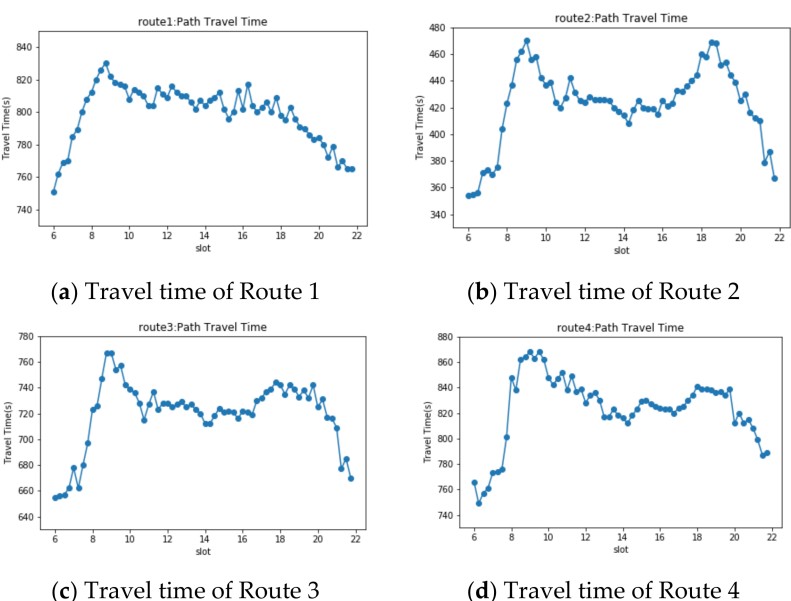

(**a**) Travel time of Route 1     (**b**) Travel time of Route 2

(**c**) Travel time of Route 3     (**d**) Travel time of Route 4

**Figure 6.** Travel time estimation line curve of four routes.

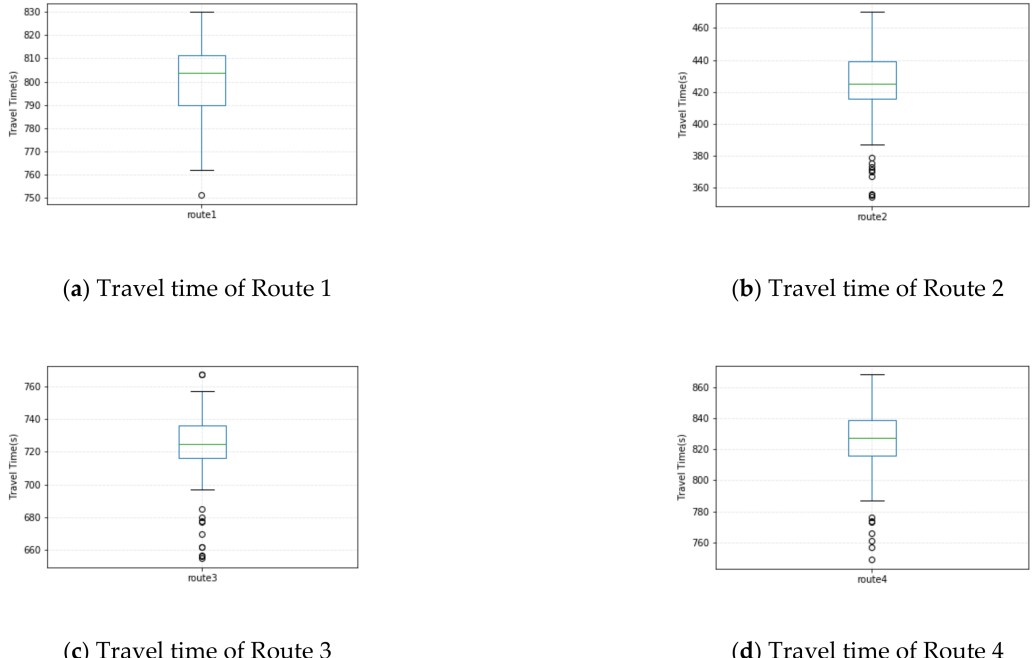

(**a**) Travel time of Route 1    (**b**) Travel time of Route 2

(**c**) Travel time of Route 3    (**d**) Travel time of Route 4

**Figure 7.** Four routes' box plot of travel time of a day prediction.

It can be seen from the box plot of the prediction results that the prediction results are scattered into the normal range. There are only some minor abnormal values. Furthermore, the abnormal values often occur in certain timeslots in the morning mostly. This phenomenon is attributed to the fact that there are fewer vehicles on the road during that time intervals. The vehicles can travel at a free-flow speed then. This results in lower delays caused by the route passing process.

The experiment tests were also conducted to compare the influence of different parameters of the algorithms. Figure 8 shows the relationship between the input length of short-term LSTM network and the prediction performance in terms of MAE. The result shows the impact of the number of short-term timeslots on the model performance. When the length of the short-term input sequence is 8, the MAE is the smallest. Therefore, the short-term input sequence length is usually chosen to be 8. Obviously, along with the sequence increasing, the computational time increases. Therefore, the response time of the algorithm increases, too, which causes the MAE to decrease first and then increase.

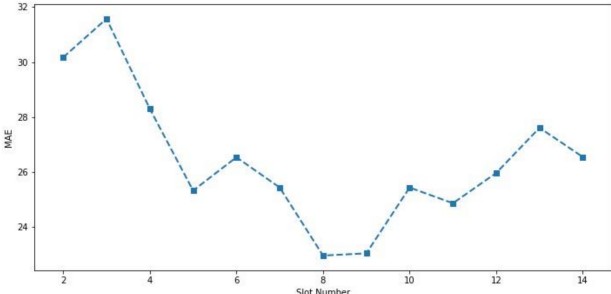

**Figure 8.** Short-term LSTM length performance in terms of MAE.

Figure 9 shows the comparison results of long-term LSTM input lengths in a number of days. As the long-term sequence length of the model increases, the model performance gradually improves. When the input length is 15, the prediction accuracy becomes gradually stable. Therefore, in the other experiments, the sequence length of the long-term LSTM algorithm is arbitrarily selected as 15 as the input value.

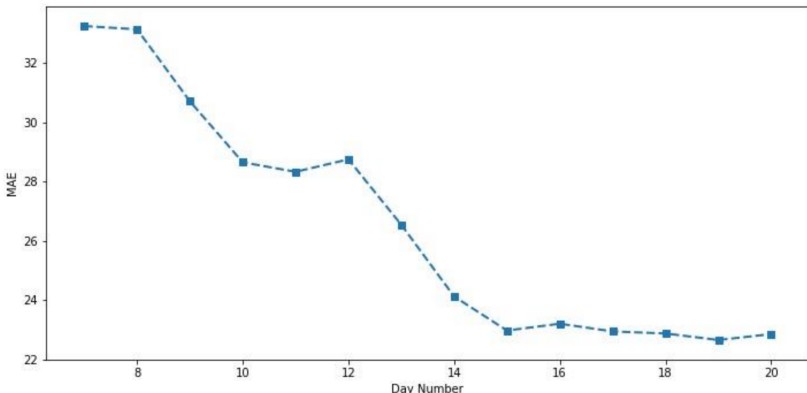

**Figure 9.** Long-term LSTM length performance in terms of MAE.

## 6. Conclusions

In this study, a novel DLSF-CNN model is developed to predict route travel time based on spatial–temporal features. The DLSF-CNN model uses the route travel time matrix to model the spatiotemporal characteristics of travel time in both temporal and spatial domains. Combining with multiple levels of deep learning models, it deals with the complex spatiotemporal dependence among routes. In fact, when the DLSF-CNN model constructs the route travel time matrix and inputs it into the convolutional neural network to capture the spatial dependence among routes, it replaces the missing values of a roadway segment of the route by the free-flow travel time. Therefore, the spatial correlation of the model is ensured. Based on the performance comparisons with the other models, the prediction accuracy of the proposed DLSF-CNN model is more effective than the other methods in terms of the average absolute error and $R^2$ score. This algorithm can predict the route travel time in the next time interval more accurately. The main contribution of the DLSF-CNN model not only improves the applicability of low-resolution sampling data for travel time prediction, but also demonstrates great potential to predict network-wide traffic status evolvement. The future work of the DLSF-CNN model is to enlarge its application scenes, such as high-resolution floating car datasets and real-time forecasting. It is not efficient enough to handle such scenes at present.

**Author Contributions:** X.L. is the main author studied on theories and the plan of experiments. H.W. is the director of this paper to handle the direction of the whole work. P.S. is the main person worked on the most part of experiments. H.Z. checked the work in detail. All authors have read and agreed to the published version of the manuscript.

**Funding:** This work has been partly funded by the European Commission within the H2020 Framework Programme under the H2020 project CONCORDIA (contract n. 830927).

**Institutional Review Board Statement:** Not applicable.

**Informed Consent Statement:** Informed consent was obtained from all subjects involved in the study.

**Data Availability Statement:** The data presented in this study are available on request from the corresponding author. The data are not publicly available due to the owner, which is Administration of Harbin Transportation.

**Conflicts of Interest:** The authors declare no conflict of interest.

## Appendix A

**Table A1.** Notations used in this paper (ordered by their appearance).

| Notation | Meaning |
|---|---|
| $L_i$ | A route segment |
| $T_i$ | A time slice of a day |
| $t_{ijk}$ | A sample from the *j*-th and *k*-th roadway segments in $T_i$ |
| $t_{ijk}$ | Timeslot travel time |
| $R_t$ | Route travel time matrix |
| $R_t^i$ | Route travel time matrix in timeslot *i* |
| $R_t^i, \mathrm{k}$ | The *k*-th route travel time matrix in timeslot *i* |
| $h_t$ | Hidden layer output |
| $f_t$ | Forgetting layer |
| $i_t$ | Input layer |
| $C_t'$ | Input layer, which is used to update the content |
| $o_t$ | Output information |
| $tanh(\cdot)$ | tanh activation function |
| $h_t^i$ | The prediction result of the partial path *i* in a route in the *t*-th timeslot |
| $h_d^{i,p}$ | The output of the convolutional neural network on the travel time along route *i* in the *d*-th timeslot of the *p*-th day |
| $\alpha_d^{i,p}$ | The weight of the attention contribution |
| $h'^{i,p}_t$ | The prediction result based on the time characteristics of coarse granularity with time shift correction by the attention mechanism-based LSTM network |
| $y_{t+1}^i$ | In the range of (–1,1), as the span of function $tanh()$ |
| $\omega_l$ | Learning parameters |
| $b$ | Learning parameters |

### Appendix B

**Table A2.** Acronyms in this paper (ordered by their appearance).

| Notation | Meaning |
|---|---|
| LSTM | Long short-term memory |
| CNN | Convolution neural network |
| DLSF | Deep learning on spatiotemporal features |
| RMSE | Root Mean Squard Error |
| $R^2$ | R Squared |
| AVI | Autonomous vehicle identification |
| GPS | Geographic position system |
| AI | Artificial intelligence |
| DAMA | Differential Autoregressive Moving Average |

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
