# Peer review of "Spatiotemporal Features—Extracted Travel Time Prediction Leveraging Deep-Learning-Enabled Graph Convolutional Neural Network Model"

_sustainability, doi:10.3390/su13031253_

Round 1
Reviewer 1 Report
The authors present their work on "Deep Learning-enabled Spatiotemporal Feature3 extracted Model Development for Travel Time Prediction" which is of interest. But still there is some work to do for the improvement.
1). The first sentence of the abstract should be revised.
2). Contribution of the paper should be highlighted in the introduction section
3). Include and discuss the deep learning approach of the following paper in the literature review.
a) Optimized gene selection and classification of cancer from microarray gene expression data using deep learning
b) Deep learning recognition of diseased and normal cell representation
4). Equation 1 need to be explained more.
5). I can't see figure (a) and (b) in figure 1
6). Figure 3 need to be explained more.
7). Figure 6 need to be sub figure with a-d
8). Section 6 need to be explained more.
Author Response
Thank you for your advising. Your advices are very valuable and important to enlighten our work. We considered these advices carefully and think over the answers or statements to write. We both believe that this paper is improved obviously.
As the comments about our paper “Deep Learning-enabled Spatiotemporal Feature-extracted Model Development for Travel Time Prediction”, which is numbered as sustainability-1061829, it has already re-edited and uploaded. For reading conveniently, the re-written sentences are following the order of the comments. If there are still some problems in this paper, please let us know and take a further study.
Thank you very much!
The authors present their work on "Deep Learning-enabled Spatiotemporal Feature extracted Model Development for Travel Time Prediction" which is of interest. But still there is some work to do for the improvement.
1). The first sentence of the abstract should be revised.
It is revised. Key work ‘prediction’ is insert into the first sentence and the whole sentence is adjusted.
2). Contribution of the paper should be highlighted in the introduction section
The contributions of this paper are highlighted in lines between 62 to 69.
3). Include and discuss the deep learning approach of the following paper in the literature review.
- a) Optimized gene selection and classification of cancer from microarray gene expression data using deep learning
- b) Deep learning recognition of diseased and normal cell representation
They are introduced in lines between 123 to 125. The citification no. of them are [18] and [19].
4). Equation 1 need to be explained more.
It is explained in lines between 151 and 153. “Here, p represents one of the m samples.”
5). I can't see figure (a) and (b) in figure 1
The title of figure 1 is re-edited. There is no sub-figure indeed.
6). Figure 3 need to be explained more.
It is explained in lines between 250 and 251. “The applicability of data from these floating cars and the action track coverage area are shown in Figure 3. It is the urban traffic network of Harbin, China.”
7). Figure 6 need to be sub figure with a-d
Sub-titles are added in the revised paper.
8). Section 6 need to be explained more.
Conclusion is deleted and refined into section 6, between lines 358 to 371.
Reviewer 2 Report
Title: Deep Learning-enabled Spatiotemporal Feature-extracted Model Development for Travel Time Prediction
Technical Comments:
- The motivation should clearly identify what drives the need for this work to be done, clear articulation of objectives of the work, and novelty of the proposed approach, impact of successful implementation of the approach. All of these are either lacking or incomplete in the manuscript
- Spatio-temporal characteristics are mentioned frequently but not explicitly defined and their importance is not identified early in the manuscript.
- Authors claim: "However, in travel time prediction, the complex nonlinear temporal and spatial dependence of partial and complete paths included in a route significantly affects its prediction accuracy" - what is this claim based on?
- A literature review is lacking in several respects. Specifically, a proper literature review of the current state of the art including limitations or gaps in existing approaches should be presented. It should:
- (a) direct future readers to relevant useful other works in the area
- (b) give the readers a perspective of the place of their work in the wider field that includes other approaches and other results in the same area
- (c) a sense of the dynamics of an evolving field: recent works, who works on what concurrently or recently, what work was done during the same era
- Why is it necessary to convert spatiotemporal data to image-type data? - The authors have not provided solid justifications for this.
- The authors have stated the research goal at the end of section 3 - but not justified how this research goal is tied to the gaps they identified in existing methods and how that goal will fill these gaps
- The authors have gone into significant detail about LSTM networks and their theory which is good, but it is also expected that they provide some better insight into their modeling choices and justifications.
- It would be beneficial to get further insights into the authors' data manipulation and processing as it seems to be very short
- The quality of images 4,5 is very poor
- The discussion section is not populated at all
Author Response
Thank you for your advising. Your advices are very valuable and important to enlighten our work. We considered these advices carefully and think over the answers or statements to write. We both believe that this paper is improved obviously.
As the comments about our paper “Deep Learning-enabled Spatiotemporal Feature-extracted Model Development for Travel Time Prediction”, which is numbered as sustainability-1061829, it has already re-edited and uploaded. For reading conveniently, the re-written sentences are following the order of the comments. If there are still some problems in this paper, please let us know and take a further study.
Thank you very much!
Technical Comments:
The motivation should clearly identify what drives the need for this work to be done, clear articulation of objectives of the work, and novelty of the proposed approach, impact of successful implementation of the approach. All of these are either lacking or incomplete in the manuscript
In the first paragraph of Introduction, the importance of accurate prediction on route travel time is pronounced. And the research target of this paper is re-edited in lines between 59 to 69.
Spatio-temporal characteristics are mentioned frequently but not explicitly defined and their importance is not identified early in the manuscript.
Actually, the problem about route travel time prediction is a typical kind of spatio-temporal problem. It is mentioned in this paper in lines 52 to 55.
Authors claim: "However, in travel time prediction, the complex nonlinear temporal and spatial dependence of partial and complete paths included in a route significantly affects its prediction accuracy" - what is this claim based on?
Route travel time cannot be regression to construct a function. It is a typical nonlinear relationship.
A literature review is lacking in several respects. Specifically, a proper literature review of the current state of the art including limitations or gaps in existing approaches should be presented. It should:
(a) direct future readers to relevant useful other works in the area
(b) give the readers a perspective of the place of their work in the wider field that includes other approaches and other results in the same area
(c) a sense of the dynamics of an evolving field: recent works, who works on what concurrently or recently, what work was done during the same era
We re-edited the introduction and related part to fulfill the points proposed above. We also added two more papers into our references by the requirements from reviewers.
Why is it necessary to convert spatiotemporal data to image-type data? - The authors have not provided solid justifications for this.
The graph-type like data is used to capture the relationship between spatial data flow. It is provided in lines 171 to 178 in the newest edition.
The authors have stated the research goal at the end of section 3 - but not justified how this research goal is tied to the gaps they identified in existing methods and how that goal will fill these gaps
It is stated in lines between 104 to 131.
The authors have gone into significant detail about LSTM networks and their theory which is good, but it is also expected that they provide some better insight into their modeling choices and justifications.
We state the LSTM networks in the whole part of section 4.2. For the reader further understanding, we collect all the characters in this paper into a table append behind.
It would be beneficial to get further insights into the authors' data manipulation and processing as it seems to be very short
It is further described in lines between 262 to 268 in the new edition.
The quality of images 4,5 is very poor
These two figures are refined.
The discussion section is not populated at all.
The discussion section is re-edited. It is shown in lines 358 to 371.
Reviewer 3 Report
Dear Authors,
Although the results presented in the manuscript seem promising and overall approach is contributing in the body of the literature, I encourage the authors to please consider the attached file concerns and suggestions. Thanks

Author Response
Thank you for your advising. Your advices are very valuable and important to enlighten our work. We considered these advices carefully and think over the answers or statements to write. We both believe that this paper is improved obviously.
As the comments about our paper “Deep Learning-enabled Spatiotemporal Feature-extracted Model Development for Travel Time Prediction”, which is numbered as sustainability-1061829, it has already re-edited and uploaded. For reading conveniently, the re-written sentences are following the order of the comments. If there are still some problems in this paper, please let us know and take a further study.
Thank you very much!
Major Comments
- Two pertinent studies were added and analyzed in lines 123 to 125.
- The structure of the model DLSF-CNN is described in figure 1. It is re-edited to be more clear.
- We were thinking about how to get realistic data and gives prediction results immediately. This is not easy, especially in a short term.
- Conclusion is deleted. The content of conclusion is re-edited and enriched discussion part.
- It is revised by reading throughout the total paper.
Minor comments
- It is revised.
- The title of this paper is named “Deep Learning-enabled Graph Convolutional Neural Network-based Model Development for Travel Time Prediction”.
- There is a new table at the end of this paper as Appendix I is about all the notations in this paper..
- R2 is corrected into the right form.
- The citations of the models were provided.
- The legend provided in Figure 5 is enlarged.
- It was deleted. The discussion part and conclusion part were combined into one part.
- Some sentences were re-edited to improve in this manuscript, such as line 39, lines 62 to 69, line 141, line 153, line 219, line 274.
Round 2
Reviewer 2 Report
Thank you for addressing many of the comments. I still think Figure 4 and 5 are very poor quality from a journal paper perspective.
Author Response
Figure 4 and figure 5 are refined and enlarge the size of fonts. Please revise it again, thanks!
Reviewer 3 Report
Dear Authors,
Although the paper has been improved substantially from the initial version, but still I feel some changes are required before it can be processed further. I encourage the authors to please consider the attached file concerns and suggestions to improve the work. Thanks

Author Response
Here are the responds to the commits. Please check it and give some further advice. Thank you very much!
Best wishes,
- Most of the titles of figures in this paper are re-edited.
- The title of this paper is adjusted mainly according to the commit 2.
- The title of figure 9 is adjusted.
- There are two references inserted into this paper, which are [18] S. H. Shah, M. J. Iqbal, I. Ahmad, S. Khan, and J. J. P. C. Rodrigues, "Optimized gene selection and classification of cancer from microarray gene expression data using deep learning," Neural Computing and Applications, p. 12, 2020/10/06 2020 and [19] M. S. Iqbal, I. Ahmad, L. Bin, S. Khan, and J. J. P. C. Rodrigues, "Deep learning recognition of diseased and normal cell representation," Trans Emerging Tel Tech., vol. n/a, no. n/a, p. e4017, 2020.
- The symbol R2 in discussion is turned into R2.
- The limitation and future study is given in lines 71 to 75.
- The abbreviation of AVI is given in the last edition.
- The sentence is corrected in the new edition.
